# The Impact of Trauma and Substance Use on Emotion Regulation and Intimate Partner Violence Perpetration: Implications for Perpetrator Programs

**DOI:** 10.3390/bs15020156

**Published:** 2025-02-01

**Authors:** Cristina Expósito-Álvarez, Manuel Roldán-Pardo, Viviana Vargas, Mina Maeda, Marisol Lila

**Affiliations:** Department of Social Psychology, Faculty of Psychology and Speech Therapy, University of Valencia, 46010 Valencia, Spain; cristina.exposito@uv.es (C.E.-Á.); manuel.roldan-pardo@uv.es (M.R.-P.); viviana.vargas@uv.es (V.V.); mima9@alumni.uv.es (M.M.)

**Keywords:** intimate partner violence, perpetrator programs, emotion regulation, trauma, alcohol and other drug use

## Abstract

(1) Background: Alcohol and/or other drug use problems (ADUPs) and trauma are key risk factors for intimate partner violence (IPV) that should be addressed in perpetrator programs. Participants with ADUPs and trauma histories may display greater difficulties in emotion regulation, which may increase the likelihood of IPV recidivism. The study aimed to examine differences among participants with trauma, ADUPs, ADUPs and trauma, and without such factors in dropout, IPV, and variables related to emotion regulation at pre- and post-intervention; (2) Methods: A sample of 312 men court-mandated to attend a perpetrator program (Contexto Program) was used. Variables related to emotion regulation difficulties included alexithymia, depressive symptomatology, and clinical syndromes. IPV variables included self-reported physical and psychological IPV and IPV recidivism risk assessed by facilitators. Comparisons between groups were made using one-way ANOVA, chi-square tests, and two-way repeated measures ANOVAs; (3) Results: Participants with ADUPs and trauma presented greater difficulties on variables related to emotion regulation, higher risk of IPV at pre-intervention, and higher dropout rates. At post-intervention, all participants improved their emotion regulation skills and reduced IPV recidivism risk, with participants with ADUPs and trauma maintaining a higher risk of IPV; (4) Conclusions: IPV perpetrators with ADUPs and trauma are high-risk participants. Interventions should target trauma and ADUPs to improve their effectiveness.

## 1. Introduction

Intimate partner violence (IPV) against women constitutes a pervasive social and public health problem of pandemic proportions, affecting approximately 1 in 3 women aged 15 to 49 years globally at least once in their lifetime ([117]). In response to the growing recognition of IPV against women as a critical social problem, intervention programs for intimate partner violence perpetrators emerged in the late 1970s as a primary strategy to prevent IPV against women by promoting attitudinal and behavioral change among men ([6]; [64]). Since then, community-based intervention programs for IPV perpetrators have become an essential element of the criminal justice system in addressing IPV, particularly for men who have committed IPV offenses and received suspended sentences contingent upon their participation in these group-based programs ([17]; [22]). However, systematic reviews have yielded mixed evidence regarding the effectiveness of perpetrator programs in reducing IPV recidivism ([4]; [80]; [100]; [115]). For instance, [80] ([80]) reported that three of the four randomized controlled trials (RCTs) included in their review demonstrated a significant reduction in IPV post-intervention, although results from non-experimental studies remained inconclusive. Similarly, the meta-analytic review conducted by [4] ([4]) observed a positive reduction in IPV recidivism based on police reports but found non-significant effects when victim-reported recidivism was considered. Furthermore, the systematic review conducted by [115] ([115]) found insufficient evidence to determine the overall effectiveness of perpetrator programs but highlighted promising intervention approaches, such as motivational strategies.

Recent meta-analyses and systematic reviews on intervention programs for IPV perpetrators suggest that program effectiveness may be enhanced by aligning interventions with the Risk–Needs–Responsivity (RNR) model ([2]) and the Principles of Effective Intervention (PEI); ([12]; [109]; [43]; [87]). These frameworks propose that optimal outcomes are achieved when interventions are tailored to the specific treatment needs of participants, using evidence-based strategies that directly address the identified criminogenic risks of perpetrators ([85]). Although these approaches offer promising benefits for developing more sensitive and responsive treatment plans for IPV perpetrators, the integration of these principles into perpetrator programs remains in its early stages ([12]; [87]). Further research is needed to refine these programs by identifying specific risk factors and key variables to target for high-risk IPV perpetrators ([111]).

Recent literature highlights alcohol and other drug use problems (ADUPs) and trauma during childhood or adolescence as prominent risk factors that require targeted intervention ([29]; [43]; [119]). Trauma, which can range from isolated incidents to complex relational experiences, involves actual or perceived psychological or physical harm ([103]). These traumatic experiences can have lasting effects, often impacting social and emotional well-being and contributing to difficulties in relationships and attachment ([52]). Traumatic experiences are notably prevalent among IPV perpetrators, with evidence indicating higher exposure to trauma in this group compared to non-violent controls ([32]; [66]). For example, [65] ([65]) found that 94% of participants in IPV perpetrator programs reported at least one-lifetime traumatic experience, and their findings indicated that post-traumatic symptoms mediated the relationship between trauma exposure and IPV perpetration. This relationship could be explained by the fact that individuals who have experienced trauma-related powerlessness may cope by asserting power and control in their current relationships, such as through controlling behaviors toward a partner ([66]). Moreover, IPV perpetrators with a history of trauma and trauma-related symptoms demonstrate poorer treatment engagement and an increased likelihood of IPV recidivism ([75]).

Concerning substance use problems, burgeoning research highlights a robust association between ADUPs and an increased likelihood of more frequent and severe IPV among perpetrators ([13]; [48]). Notably, participants with ADUPs comprise approximately 50% of all participants in perpetrator programs ([19]; [45]). This group of participants has been identified as high-risk and highly resistant perpetrators because they present higher dropout and recidivism rates ([42]; [57]; [83]; [108]). They also exhibit multiple risk factors at multiple levels, including individual factors (e.g., depression, anxiety), relational factors (e.g., poor intimate social support), and attitudinal factors (e.g., attributing responsibility for their violent behavior to personal circumstances; ([16]; [28], [31]; [78])), that require attention in perpetrator programs ([53]). The literature has also consistently shown that IPV perpetrators with ADUPs have a significantly higher incidence of childhood trauma exposure ([1]; [47]; [71]; [88]; [93]; [96]; [108]; [110]). In this line, [34] ([34]) identified a pathway into substance use-related IPV, in which perpetrators reported using alcohol and other drugs as a coping strategy to manage emotional distress and mental health issues stemming from unresolved past trauma. Thus, understanding the emotional dynamics of IPV perpetrators may be critical for developing effective interventions while targeting important risk factors for IPV perpetration such as trauma and ADUPs.

Emotion regulation plays a pivotal role in IPV perpetration, impacting it both directly ([67]; [95]), and indirectly through its association with related risk factors such as ADUPs ([114]). Emotion regulation encompasses a range of processes through which individuals manage and respond to their emotional experiences ([39]). According to [37]’s ([37]) framework, emotion regulation includes emotional awareness, the ability to identify and describe feelings, emotional understanding and acceptance, as well as impulse control and the use of strategies to effectively modulate emotional responses. In IPV perpetrators, a limited ability to identify and express inner emotions (i.e., alexithymia), combined with other risk factors such as anxiety, depression, or trauma symptoms, may lead to emotion regulation difficulties that increase the likelihood of violent behavior ([90]; [99]). This effect may be particularly significant for high-risk individuals, especially those with a history of trauma and/or ADUPs ([93]). IPV perpetrators with alexithymia traits, who struggle to identify, describe, and express their feelings, may also exhibit maladaptive emotion regulation strategies ([73]; [91]). These maladaptive emotion regulation strategies, which include rumination, substance use, and experiential avoidance, have been associated with elevated anxiety levels, persistent sadness (i.e., dysthymia), and post-traumatic and depressive symptoms ([33]; [118]). For instance, using alcohol as a maladaptive emotion regulation strategy may intensify emotional instability, contributing to perpetuating the cycle of violence in IPV perpetrators ([38]). Conversely, developing emotion regulation skills, such as cognitive reappraisal, seeking support, and emotional awareness, can help individuals manage difficult emotions, such as post-traumatic symptoms resulting from previous trauma ([7]; [59]). In addition, improving emotion regulation skills could help perpetrators reduce their risk of recidivism ([67]). Considering the strong correlation between IPV perpetration and emotion regulation difficulties, such as anxiety, depression, dysthymia, and post-traumatic symptoms ([99]), it remains crucial to assess these variables in IPV perpetrators with ADUPs and/or trauma histories. In addition, intervention programs for IPV perpetrators should incorporate evidence-based emotion regulation strategies as a core component of the program structure for those in need ([68]). For example, a study conducted by [9] ([9]) that participants assigned to a trauma-informed group intervention designed to reduce IPV in veterans, the Strength at Home program ([105]), demonstrated significantly greater reductions in alexithymia and IPV than an enhanced treatment as usual for veterans. Similarly, another study showed that the Strength at Home program was significantly associated with reductions in physical and psychological IPV and post-traumatic stress disorder symptoms in veterans ([20]). While this body of knowledge highlights the importance of addressing trauma to reduce IPV and emotion dysregulation in veterans, more research is needed to evaluate such outcomes in intervention programs for men convicted of IPV ([9]; [20]; [105]).

This study aimed to assess variables related to emotion regulation at pre- and post-intervention among men court-mandated to attend a community-based program for IPV perpetrators, comparing those with trauma and/or ADUPs to those without these risk factors to explore the potential need for specialized interventions. Additionally, this study sought to examine the risk of recidivism assessed by program facilitators, participants’ self-reported IPV, and dropout rates among these participants. The comparison groups were selected based on prior research identifying trauma and substance use as critical factors influencing IPV and emotion regulation ([13]; [93]; [108]). Variables related to emotion regulation difficulties included alexithymia, depressive symptomatology, and clinical syndromes, such as anxiety, dysthymia, and post-traumatic stress disorder. These constructs, which are closely linked to a higher risk of IPV perpetration ([99]), were included as they capture challenges in identifying, managing, and responding to emotional experiences, which are central to emotion regulation processes ([37]; [89], [90]). We hypothesized that participants with ADUPs and trauma would exhibit greater difficulties with variables related to emotion regulation, higher IPV perpetration, and an increased risk of recidivism and dropout compared to those without these risk factors.

## 2. Materials and Methods

### 2.1. Sample

The sample consisted of 312 male participants who were court-ordered to participate in an intervention program for IPV perpetrators (Contexto Program, ([56])). The participants were enrolled in one of the 29 different intervention groups that were developed between June 2019 and September 2023. The following criteria were used to determine eligibility: (a) male aged 18 years or above; (b) a conviction for IPV against a current or former partner; (c) the absence of a serious mental disorder; (d) completion of the assessment and motivational phase; and (e) signed the informed consent. The mean age of the participants was 41 years (*SD* = 11.3; 19–79). The majority of participants (77.5%) were from Spain, 9.9% were from other European countries, 7.3% were from a Latin American country, 4.7% were from an African country, and 0.6% were from an Asian country. With regard to marital status, 77.2% of the sample were single, separated, divorced, or widowed, while the remaining 22.8% reported that they were married or in a partnership. In accordance with their educational level, 8.4% of the participants had no studies, 47.9% had completed elementary studies, 36% had completed high school, and 7.7% had obtained a college degree. A total of 37.2% of the sample were unemployed, and the median annual family household income fell between EUR 6000 and EUR 12,000. Table 1 provides a detailed overview of the socio-demographic characteristics of the sample.

### 2.2. Contexto Program

The Contexto Program ([56]) is a community-based intervention for male perpetrators of IPV, implemented at the University of Valencia (Spain). The program comprises 35 weekly sessions, each lasting two hours, totaling 70 h over approximately 11 months. The intervention is delivered in a group format, with each group consisting of 10 to 12 participants, and facilitated by two trained psychologists. The groups operate as closed cohorts, meaning that once the intervention commences, no new participants are permitted to join.

The intervention is guided by a theoretical framework that integrates the cognitive-behavioral theory, feminist approach, and the ecological model, in alignment with the World Health Organization’s recommendations ([116]). The group-based intervention comprises five modules ([55]). Specifically, module two is designed to enhance skills in emotional identification, regulation, and expression, as well as self-control. This module employs techniques such as cognitive restructuring and anger management techniques. Additionally, the intervention incorporates the Individualized Motivational Plan (IMP; ([56])). The IMP comprises a series of motivational strategies designed to enhance treatment compliance and foster motivation to change. The IMP is grounded in motivational interviewing ([76]), stages of change approach ([84]), solution-focused brief therapy ([24]), the Good Lives Model ([49]), and therapeutic alliance ([11]). The IMP is comprised of four primary strategies. First, five individual motivational interviews are conducted (with each interview lasting approximately 1 hour) to identify personal goals related to IPV and to monitor their achievement. Second, three group sessions are held to share progress and achievements toward goals and receive feedback and support from the group. Third, the therapist reinforces participants’ goals in the weekly group sessions. Fourth, retention techniques are also included throughout the intervention process.

### 2.3. Instruments

Trauma. The Spousal Assault Risk Assessment Guide (SARA; ([46]); Spanish version by [3] ([3]); for a full description of the SARA see the information below) was used to assess participants’ exposure to trauma during childhood or adolescence. Only item 6 (victim of and/or witness to family violence as a child or adolescent) was considered. Responses were solicited on a 3-point Likert-type scale. Responses were dichotomized into 0 (absence of trauma history) and 1 (presence of trauma history).

Alcohol and Other Drug Use Problems. The alcohol dependence and/or substance dependence scales of the Millon Clinical Multiaxial Inventory-III (MCMI-III; ([77]); Spanish version by [15] ([15]); for a full description of the MCMI-III see the information below) were used to assess the participants’ current/past alcohol and/or other drugs use problems. A score of 75 or above on one or both subscales was indicative of ADUPs. Cronbach’s alpha reliability coefficients for both alcohol dependence and substance dependence were found to be 0.71 and 0.80, respectively ([15]).

#### 2.3.1. Dropout and Intimate Partner Violence

*Dropout.* The participants were assigned a score of 0 if they completed the intervention program and a score of 1 if they ceased attendance. In this context, “ceased attendance” specifically refers to dropout from the perpetrator program, rather than discontinuation of participation in the study.

Self-reported IPV. The extent to which individuals employed violent behaviors against their partner in the past 12 months was assessed using the self-reported physical and psychological violence subscales from The Revised Conflict Tactics Scale (CTS-2); ([102]); Spanish version by [63] ([63]). Responses were on an 8-point Likert-type scale, with values ranging from 0 = this has never happened to 6 = more than 20 times in the past year, and 7 = never in the past 12 months, but it has happened before. The frequency-based scoring method proposed by [102] ([102]) was employed. To mitigate the impact of asymmetric and skewed distributions, extreme outliers were subjected to truncation. This method was originally outlined by [98] ([98]) and has since been employed in analogous studies ([29]; [51]; [69]). The CTS-2 has demonstrated construct and discriminant validity ([102]), and its Spanish version has been extensively administered to samples of IPV male perpetrators ([56]; [63]). The Cronbach alpha reliability coefficients for physical and psychological violence were 0.60 and 0.80, respectively, at the pre-intervention stage, and 0.86 and 0.72, respectively, at the post-intervention stage.

Risk of Recidivism Assessed by the Facilitators. The SARA ([3]; [46]) was used by the facilitators to assess the risk of recidivism. The protocol employs a 20-item clinical checklist format that includes the primary risk factors for IPV to indicate the risk of recidivism. Responses were provided on a 3-point Likert-type scale to ascertain the presence or absence of each risk factor (0 = absent, 1 = possibly present, 2 = present). The overall score was employed as an indicator of the total risk of recidivism. A higher score was indicative of a greater risk of recidivism. SARA has proven to have predictive validity ([74]), and the Spanish version of the instrument has been utilized in samples of IPV male perpetrators ([56]; [62]).

#### 2.3.2. Variables Related to Emotion Regulation Difficulties

Alexithymia. The Toronto Alexithymia Scale—20 Items (TAS-20); ([8]); Spanish version by [70] ([70]), was employed to assess the trait of alexithymia. The TAS-20 comprises 20 items presented on a 6-point Likert-type scale, ranging from 1 (strongly disagree) to 6 (strongly agree). In this study, only two subscales were used: difficulty identifying feelings and difficulty describing feelings. The TAS-20 has exhibited satisfactory reliability and validity ([8]), and the Spanish version has been employed in samples of IPV perpetrators ([18]; [89]). Cronbach’s alpha reliability coefficients for difficulty identifying feelings and difficulty describing feelings subscales were 0.89 and 0.75, respectively, at pre-intervention, and 0.90 and 0.74, respectively, at post-intervention.

Depressive symptomatology. The Center for Epidemiologic Studies Depression Scale (CES-D); ([86]); a Spanish short version by [41] ([41]), was used to evaluate the participants’ frequency and severity of depressive symptomatology in the past week. The scale includes 7 items based on a 4-point Likert-type scale, ranging from 1 (*rarely*) to 4 (*all the time or most of the time*). The Spanish short version has been previously used with samples of IPV male perpetrators ([27]; [58]). In this study, Cronbach’s alpha coefficient was 0.89 at pre-intervention and 0.88 at post-intervention.

Clinical syndromes. The MCMI-III ([15]; [77]) was employed to assess the presence of moderate clinical syndromes. The self-report inventory comprises 175 true-false questions, and only the following scales were utilized: anxiety, dysthymia, and post-traumatic stress disorder. A score of 75 or above is indicative of the presence of a significant clinical syndrome. The Spanish version has been used in samples of IPV male perpetrators ([14]; [96]). The Cronbach’s alpha reliability coefficients for the anxiety, dysthymia, and post-traumatic stress disorder scales were 0.83, 0.87, and 0.86, respectively ([15]).

### 2.4. Procedure

Data on socio-demographic characteristics, self-reported IPV, alexithymia, and depressive symptomatology were collected as a part of the initial assessment for participants attending the intervention program and at the conclusion of the intervention. The aforementioned data were collected using a self-report assessment battery, which was administered by program staff in three two-hour assessment sessions: two during the assessment and motivation phase prior to the intervention program, and one after the conclusion of the intervention. Data on clinical syndromes were collected exclusively at pre-intervention. Data on the risk of recidivism were evaluated by the facilitators after the assessment and motivation phase, and at the conclusion of the intervention. Finally, data on dropout rates were collected at the conclusion of the intervention. A total of 84 participants dropped out from the intervention for the following reasons: reincarceration (*n* = 12; 14.3%), absenteeism (*n* = 52; 61.9%), health problems (*n* = 12; 14.3%), and highly disruptive behavior during the intervention (*n* = 8; 9.5%). During the assessment and motivation phase, written informed consent was requested from the participants. The participants were informed that neither their decision to participate nor their decision to decline participation in the present study would affect their legal status or provide any legal benefit. All data were collected in accordance with approved procedures by the Ethics Committee of the University of Valencia (H1537520365110).

When all the data were collected, the participants were classified into four groups: *ADUPs*, *Trauma*, *ADUPs and trauma*, and no history of ADUPs and trauma (None). Participants were classified in the *ADUPs* group (*n* = 79) if they achieved a score of 75 or above on the alcohol dependence and/or substance dependence scales of the MCMI-III ([77]). Participants were classified in the *Trauma* group (*n* = 61) if they scored greater than 0 on item 6 of the SARA ([3]). Participants were classified in the *ADUPs and trauma* group (*n* = 66) if they both scored 75 or above on the alcohol dependence and/or substance dependence scales of the MCMI-III ([77]) and a score higher than 0 on item 6 of the SARA ([3]). Participants were classified into the *None* group (*n* = 106) if they did not achieve any of the aforementioned scores.

### 2.5. Data Analysis

First, the comparability of participants across all groups during the adjudication period was examined. In order to compare categorical and continuous variables between groups, chi-square test and one-way ANOVA, respectively, were conducted. The Kruskal–Wallis test was employed for continuous variables with a non-normal distribution. Sociodemographic characteristics (e.g., age, annual income, country of origin, marital status, educational level, employment) were included.

Second, one-way ANOVAs were conducted to evaluate the differences between the groups at pre-intervention. The normality assumption was evaluated through the implementation of the Shapiro–Wilk test and the homogeneity assumption was evaluated through Levene’s test. HSD post-hoc test was used to assess the differences between groups ([79]). The effect size was calculated using the eta-squared statistic. The multiple imputation method (MI) was employed for the handling of missing data if there was any. Despite the inherent limitations of any imputation method used to address missing data ([107]), MI, by fully conditional specification, is regarded as a viable approach for the management of missing data in both categorical and continuous variables ([61]). The variables included were self-reported IPV, risk of recidivism, alexithymia, depressive symptomatology, and clinical syndromes.

Finally, several analyses were performed to assess the differences between the groups after the end of the intervention. Chi-square tests were employed for categorical variables, whereas two-way repeated measures ANOVAs were conducted for continuous variables. The sphericity assumption was evaluated using Mauchly’s test of sphericity. In instances where sphericity was not assumed, corrected tests were employed. The Huynh-Feldt correction was employed when the Epsilon value exceeded 0.75, while the Greenhouse-Geisser correction was employed when the epsilon value was less than 0.75 ([5]). To assess the differences between groups and over time, Tuckey’s HSD post hoc test was used ([79]). The effect size was calculated using Cramér’s *V* statistic for categorical variables and the eta-squared statistic for continuous variables. MI was applied to manage any potential missing data. The variables included were dropout, self-reported IPV, risk of recidivism, alexithymia, and depressive symptomatology. The data set utilized in the present study has not been employed in previous research. All statistical analyses were performed using the IBM SPSS Statistics software, version 28.0.1.1.

## 3. Results

### 3.1. Baseline Characteristics

Table 2 presents a comparison of the baseline characteristics of the participants in each of the groups. Six variables were examined across the four groups, and none of the comparisons reached statistical significance (*p* > 0.05). The findings indicated that the groups were statistically comparable in terms of the baseline characteristics.

### 3.2. Pre-Treatment

#### 3.2.1. Dropout and Intimate Partner Violence

Table 3 displays the descriptive statistics for each group of variables analyzed at pre-intervention. The results indicated a significant difference between groups in the risk of IPV recidivism (*F*_1, 294_ = 33.1, *p* < 0.001, *η*^2^*_p_* = 0.26), self-reported physical violence (*F*_1, 308_ = 6.51, *p* < 0.001, *η*^2^*_p_* = 0.06), and psychological violence (*F*_1, 309_ = 7.05, *p* < 0.001, *η*^2^*_p_* = 0.07). Specifically, participants in the *ADUPs and trauma* group exhibited a greater risk of IPV recidivism and self-reported physical violence compared to those in the *Trauma* (*ΔM* = 2.79, *p* = 0.004; *ΔM* = 4.06, *p* = 0.002, respectively), *ADUPs* (*ΔM* = 3.92, *p* < 0.001; *ΔM* = 3.33, *p* = 0.008; respectively), and *None* (*ΔM* = 6.75, *p* < 0.001; *ΔM* = 3.96, *p* < 0.001, respectively) groups. Additionally, these participants reported higher rates of psychological violence when compared to those in the *ADUPs* (*ΔM* = 4.36, *p* = 0.040), and *None* (*ΔM* = 7.08, *p* < 0.001) groups. Furthermore, participants in the *ADUPs* group and *Trauma* group exhibited a greater risk of IPV recidivism compared to those in the *None* group (*ΔM* = 2.83, *p* < 0.001; *ΔM* = 2.83, *p* < 0.001, respectively). No statistically significant differences were obtained for the remaining comparisons (*p* > 0.05).

#### 3.2.2. Variables Related to Emotion Regulation Difficulties

Table 3 presents the descriptive statistics for each group of the variables under examination at pre-intervention. The findings showed a significant difference in depressive symptomatology (*F*_1, 310_ = 6.67, *p* < 0.001, *η*^2^*_p_* = 0.06), difficulties in identifying and describing feelings (*F*_1, 310_ = 8.01, *p* < 0.001, *η*^2^*_p_* = 0.07; *F*_1, 310_ = 4.55, *p* = 0.004, *η*^2^*_p_* = 0.04, respectively), anxiety (*F*_1, 290_ = 13.7, *p* < 0.001, *η*^2^*_p_* = 0.13), dysthymia (*F*_1, 290_ = 10.1, *p* < 0.001, *η*^2^*_p_* = 0.09), and post-traumatic stress (*F*_1, 290_ = 18.8, *p* < 0.001, *η*^2^*_p_* = 0.16) between the groups. In particular, participants in the *ADUPs and trauma* group exhibited higher levels of depressive symptomatology (*ΔM* = 0.53, *p* < 0.001), difficulties in identifying and describing feelings (*ΔM* = 0.91, *p* < 0.001; *ΔM* = 0.69, *p* = 0.001, respectively), anxiety (*ΔM* = 33.7, *p* < 0.001), dysthymia (*ΔM* = 24.4, *p* < 0.001), and post-traumatic stress (*ΔM* = 31.7, *p* < 0.001), when compared to participants in the *None* group. Additionally, these participants presented greater difficulties in identifying feelings (*ΔM* = 0.21, *p* = 0.035) and anxiety levels (*ΔM* = 17.6, *p* = 0.024), when compared to those in the *Trauma* group, and higher prevalence of dysthymia and post-traumatic stress (*ΔM* = 12.5, *p* = 0.045; *ΔM* = 14.7, *p* = 0.009, respectively), when compared to those in the *ADUPs* group. Finally, participants in both the *ADUPs* group and *Trauma* group exhibited greater levels of depressive symptomatology (*ΔM* = 0.35, *p* = 0.023; *ΔM* = 0.40, *p* = 0.014; respectively), anxiety (*ΔM* = 18.9, *p* = 0.001; *ΔM* = 16.1, *p* = 0.021; respectively), dysthymia (*ΔM* = 11.8, *p* = 0.026; *ΔM* = 11.9, *p* = 0.049; respectively), and post-traumatic stress (*ΔM* = 17.0, *p* < 0.001; *ΔM* = 18.9, *p* < 0.001; respectively), when compared to those in the *None* group. The remaining comparisons yielded no statistically significant results (*p* > 0.05).

### 3.3. Post-Treatment

#### 3.3.1. Dropout and Intimate Partner Violence

Table 4 displays the descriptive statistics of the variables examined within each group after the end of the intervention. In general, participants showed a decrease in the risk of IPV recidivism (*F*_1, 214_ = 166.2, *p* < 0.001, *η*^2^*_p_* = 0.44) and psychological violence (*F*_1, 216_ = 12.3, *p* < 0.001, *η*^2^*_p_* = 0.05) at the end of the intervention. Furthermore, the results indicated significant differences in the risk of IPV recidivism (*F*_1, 214_ = 7.61, *p* < 0.001, *η*^2^*_p_* = 0.10) and psychological violence (*F*_1, 216_ = 3.95, *p* = 0.009, *η*^2^*_p_* = 0.05) across groups and over time. Specifically, participants in the *ADUPs and trauma* group showed a higher risk of IPV recidivism compared to the participants in the *ADUPs* (*ΔM* = 3.37, *p* < 0.001), *Trauma* (*ΔM* = 2.52, *p* = 0.005), and *None* (*ΔM* = 5.58, *p* < 0.001) groups. Additionally, these participants reported higher levels of psychological violence compared to those in the *ADUPs* (*ΔM* = 5.12, *p* = 0.009), and *None* (*ΔM* = 6.55, *p* < 0.001) groups. Moreover, participants in the *ADUPs* group and *Trauma* group exhibited a higher risk of IPV when compared to those participants in the *None* group (*ΔM* = 2.21, *p* < 0.001; *ΔM* = 3.05, *p* < 0.001, respectively). Furthermore, regarding dropout rates, participants in the *ADUPs* and *Trauma* group exhibited a higher dropout rate from the intervention than participants in the *None* group (*χ*^2^ = 8.63, *p* = 0.035, *V* = 0.17). The remaining comparisons and time effects were not statistically significant (*p* > 0.05). For a graphical representation of these results, see Figure 1.

#### 3.3.2. Variables Related to Emotion Regulation

Table 4 presents the descriptive statistics for each group of the variables under examination after the end of the intervention. Overall, participants exhibited a reduction in depressive symptomatology (*F*_1, 216_ = 16.9, *p* < 0.001, *η*^2^*_p_* = 0.07), difficulties in identifying feelings (*F*_1, 216_ = 12.9, *p* < 0.001, *η*^2^*_p_* = 0.06), and difficulties describing feelings (*F*_1, 216_ = 4.85, *p* = 0.029, *η*^2^*_p_* = 0.02) at the end of the intervention. Non-significant differences between groups were found for variables related to emotion regulation at the end of the intervention (*p* > 0.05). For a graphical representation of these results, see Figure 2.

## 4. Discussion

This study aimed to evaluate dropout rates, IPV, and variables related to emotion regulation difficulties among IPV perpetrators with trauma and/or ADUPs or without these risk factors, at the beginning of a perpetrator program and at program completion. Specifically, participants were screened against ADUPs and trauma and classified, within the context of this study, into (1) *ADUPs and trauma* group, (2) *Trauma* group, (3) *ADUPs* group, and (4) *None* group.

At pre-intervention, participants in the *ADUPs and trauma* group demonstrated a greater risk of IPV recidivism and self-reported physical violence compared to those in the *Trauma*, *ADUPs*, and *None* groups. Additionally, participants in the *ADUPs* group and *Trauma* group exhibited a greater risk of IPV recidivism compared to those in the *None* group. These results support the view that IPV perpetrators with ADUPs and trauma are high-risk, consistent with recent research that underscores the role of trauma and early exposure to violence in perpetuating subsequent IPV perpetration ([101]; [110]; [113]). Furthermore, research establishes a clear link between ADUPs and increased IPV perpetration ([23]; [48]; [53]). Additionally, unresolved trauma may exacerbate ADUPs, consistent with the self-medication hypothesis ([44]), which states that alcohol and drug consumption may be used as a coping mechanism to alleviate difficult emotions associated with trauma among IPV perpetrators ([50]). Consequently, addressing underlying trauma in IPV perpetrators with histories of substance abuse is necessary for perpetrator programs to improve participants’ outcomes ([54]; [108]).

Concerning self-reported psychological violence, participants in the *ADUPs and trauma* group reported significantly higher levels than those in the *ADUPs* and *None* group. This finding is consistent with research showing a significant association between traumatic experiences and psychological IPV perpetration ([65]; [94]). Specifically, research suggests that men with early traumatic experiences may have a heightened tendency to perceive threat, making relationship conflicts potential trauma triggers that amplify dysregulated responses and increase the likelihood of both substance use and psychological IPV perpetration ([38]; [47]; [112], [113]).

Regarding IPV at post-intervention, participants showed a decrease in the risk of IPV recidivism and psychological violence at the end of the intervention, regardless of their ADUPs or trauma screening status. However, while our results demonstrate that the program effectively reduces IPV recidivism risk among participants, they also indicate that the presence of trauma and/or ADUPs in perpetrators continues to pose an elevated risk to victims, underscoring the urgent need to address these factors ([35]; [43]; [72]). Our results also indicated that dropout rates were significantly higher for participants in the *ADUPs and trauma* group compared to those in the *None* group. This is consistent with prior research showing that men attending intervention programs for IPV perpetrators who did not experience potentially traumatic events during their past had a significantly higher probability of completing the program than those with trauma histories ([21]).

Regarding variables related to emotion regulation difficulties at pre-intervention, overall, participants with ADUPs and trauma exhibited lower emotion regulation levels across variables compared to participants without such risk factors. Specifically, participants in the *ADUPs and trauma* group demonstrated heightened difficulties in identifying and describing feelings compared to those with neither ADUPs nor trauma exposure. These emotion regulation-related challenges suggest that individuals in the *ADUPs and trauma* group may experience some deficits in emotional awareness and a reduced propensity for reflecting on their inner emotional states ([67]; [96]). Moreover, this group displayed significantly higher levels of depressive symptomatology, anxiety, dysthymia, and post-traumatic stress relative to participants in the *None* group, aligning with prior research linking ADUPs and trauma to greater psychological distress ([1]; [93]). Moreover, when comparing the *ADUPs and trauma* group to the *trauma* group, participants with co-occurring ADUPs and trauma demonstrated greater difficulties in identifying feelings and higher anxiety levels, suggesting a cumulative effect of trauma and substance use on emotional dysregulation and anxiety symptoms ([10]; [28], [31]; [97]). Furthermore, participants in the *ADUPs and trauma* group had a higher prevalence of dysthymia and post-traumatic stress compared to those in the *ADUPs* group, underscoring the potential for trauma exposure to exacerbate depression and stress-related symptoms in individuals with substance use problems ([50]; [93]).

At post-intervention, participants demonstrated a reduction in depressive symptomatology, as well as difficulties in identifying feelings and describing feelings, regardless of their ADUPs or trauma screening status. These findings could indicate that the program was effective in enhancing those emotion regulation skills among participants who completed the intervention. However, given the higher dropout rate among participants with ADUPs and trauma, it is essential to implement strategies that enhance treatment adherence in this high-risk group. These strategies may include motivational approaches to strengthen working alliances ([106]), goal setting tailored to participants’ specific needs ([27]), and targeted retention techniques, all aimed at ensuring these individuals fully benefit from the intervention’s positive effects ([57]).

Our results suggest the presence of additional important treatment implications. First, this study showed that IPV perpetrators do not constitute a homogeneous group, but rather, they have different risk levels and possess specific characteristics (i.e., trauma and ADUP histories) that should be evaluated and addressed, moving interventions beyond the “one-size-fits-all” model to more individualized approaches ([7]; [12]; [92]; [109]). This idea echoes the RNR ([2]) and PEI ([87]) approaches which stand for adjusting the interventions to participants’ risks and needs through evidence-based strategies. Specifically, and consistent with our hypothesis, our results showed that participants in the *ADUPs and trauma* group were at higher risk of IPV, experienced greater difficulties with emotion regulation variables, and had a higher likelihood of dropout than those without those factors. Our findings replicate prior evidence on the links between trauma, substance use, IPV, and emotion dysregulation ([25]; [40]; [104]; [108]). However, this study advances the field by identifying critical areas for tailored intervention in programs for IPV perpetrators, which often fail to integrate trauma-informed and targeted strategies ([109]; [111]). Therefore, our findings may help inform program facilitators on the importance of implementing trauma-informed and IPV-ADUPs integrated strategies to reduce the likelihood of recidivism and enhance victim’s safety ([26]; [110]). For instance, trauma-informed approaches that emphasize the safe disclosure of trauma histories in a group setting, the reframing of past experiences, and the enhancement of awareness around unmet emotional needs may improve participants’ emotion regulation and trauma recovery, which, in turn, may help reduce IPV recidivism ([34]). Moreover, a recent randomized controlled trial demonstrated that the incorporation of individualized motivational plans tailored to the specific needs of participants with ADUPs was effective in reducing alcohol use, increasing program engagement, and advancing participants’ stages of change compared to standard motivational strategies ([29]). In a similar vein, implementing goal setting as a core collaborative motivational strategy has proven effective in reducing dropout rates among IPV perpetrators with ADUPs, which is a key protective factor against IPV recidivism ([27], [30]). Overall, incorporating motivational-based substance use interventions alongside a trauma-informed approach shows promising results compared to standard perpetrator programs ([43]; [72]; [111]). However, the implementation of these targeted approaches into perpetrator programs is only just emerging, and further intervention efforts should be made ([87]).

Second, individuals with a history of trauma may use alcohol or drugs to regain a sense of control over their lives, which can exacerbate their emotion regulation difficulties and increase the likelihood of interpersonal conflicts ([34]). However, further research should be conducted to evaluate the mechanisms underlying the relationships between trauma, ADUPs, and IPV. Moreover, specific evidence-based strategies such as enhanced anger management techniques should be implemented for those participants who require them ([81]). Further research efforts should prioritize the evaluation of the participants’ risk factors, the development of tailored intervention protocols, and the evaluation of their effectiveness.

This study has certain limitations. While the participants’ trauma history was assessed by facilitators using the SARA ([46]) and based on the information gathered during the three individual motivational interviews, incorporating additional validated measures could improve trauma assessment. Future research should prioritize the development and validation of comprehensive measures that not only evaluate posttraumatic symptomatology but also assess the presence of trauma history, particularly complex and relational trauma, given their potentially significant impact on IPV perpetration ([69]). Post-traumatic growth should also be evaluated to assess the transformative process in men following trauma-informed interventions aimed at increasing acknowledgment of their violent behavior and fostering efforts to repair the harm inflicted on victims ([82]). While constructs relevant to emotion regulation difficulties were investigated, this study lacked a questionnaire specifically focused on emotional dysregulation, such as the Difficulties in Emotion Regulation Scale (DERS); ([37]). Future studies would also benefit from incorporating official recidivism rates into their analyses, as self-reported data may be influenced by participants’ desire to avoid self-incrimination or provide socially desirable responses. To mitigate this limitation, the present study included a risk assessment measure completed by program facilitators.

Moreover, post-intervention measures, such as variables related to emotion regulation and risk of recidivism, were only obtained for those participants who completed the intervention. Future research should account for missing data from dropouts, considering the well-established relationship between dropout rates and IPV recidivism ([58]; [83]). Furthermore, our findings are specific to men court-mandated to attend intervention programs for IPV perpetrators and may not be generalizable to other populations, such as men with prior criminal records (e.g., sexual offenses), imprisoned men, or men in non-heterosexual relationships ([36]; [60]).

Nevertheless, this study demonstrated several strengths. Emotion regulation was assessed using several variables that encompass different aspects of emotional awareness and emotional management relevant to IPV perpetrators, including alexithymia, dysthymia, depression, anxiety, and post-traumatic symptoms ([68]; [89]). Notably, to the best of our knowledge, this is the first study to evaluate the specific treatment needs and risk factors of IPV perpetrators with both trauma and ADUPs, compared to those with either trauma or ADUPs alone, as well as those without these risk factors. Our results may serve to inform researchers, professionals, and policymakers about the significance of evaluating specific risk factors among IPV perpetrators, such as trauma and substance use. They also underscore the urgent need for targeted perpetrator programs to address these factors as a critical strategy for reducing IPV recidivism and protecting victims.

## Figures and Tables

**Figure 1 behavsci-15-00156-f001:**
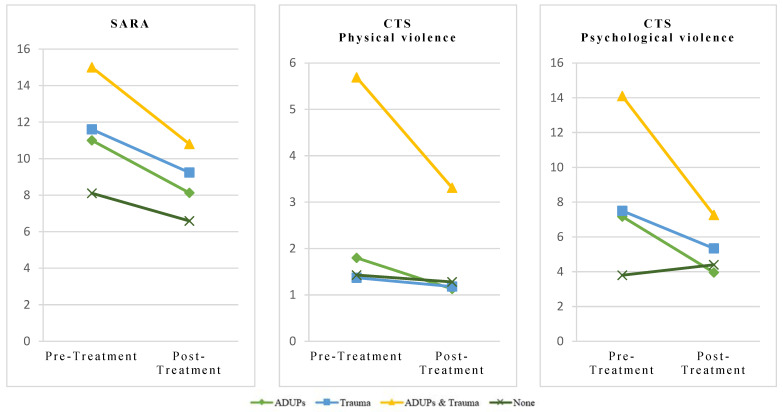
Comparison of the risk of intimate partner violence over time by group.

**Figure 2 behavsci-15-00156-f002:**
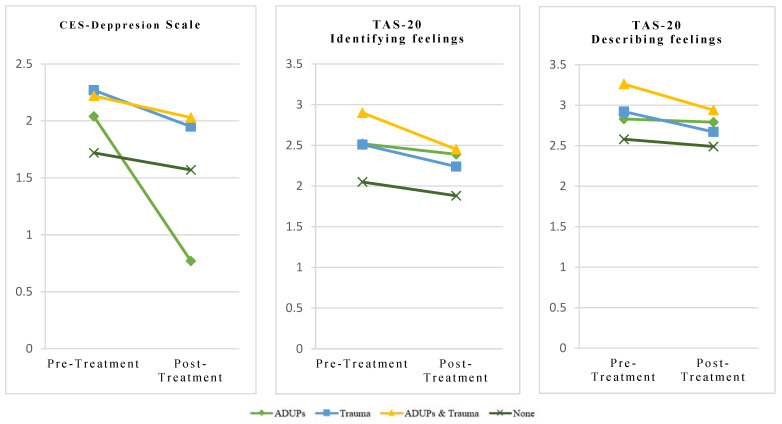
Comparison of variables related to emotion regulation over time by group.

**Table 1 behavsci-15-00156-t001:** Socio-demographic characteristics of the sample.

Variables	*M* (*SD*)	Range	*n* (*%*)
Age	41 (11.3)	19–79	
Annual income ^1^	4.58 (2.33)	1–12	
Origin			
Spain			242 (77.5)
Latin America			23 (7.3)
Europe (except Spain)			31 (9.9)
Africa			14 (4.7)
Asia			2 (0.6)
Level of education			
No education			26 (8.4)
Elementary			150 (47.9)
High School			112 (36)
College			24 (7.7)
Marital status			
Married or with partner			71 (22.8)
Single			119 (38.1)
Separated			34 (10.9)
Divorced			86 (27.6)
Widowed			2 (0.6)
Employed			
Yes			196 (62.8)
No			116 (37.2)

*Note. M* = Mean, *SD* = Standard Deviation. ^1^ Annual income: 1 = <EUR 1800, 2 = EUR 1800–EUR 3600, 3 = EUR 3600–EUR 6000, 4 = EUR 6000–EUR 12,000, 5 = EUR 12,000–EUR 18,000, 6 = EUR 18,000–EUR 24,000, 7 = EUR 24,000–EUR 30,000, 8 = EUR 30,000–EUR 36,000, 9 = EUR 36,000–EUR 60,000, 10 = EUR 60,000–EUR 90,000, 11 = EUR 90,000–EUR 120,000, and 12 = >EUR 120,000.

**Table 2 behavsci-15-00156-t002:** Baseline characteristics across groups.

Variables	ADUPs (*n* = 79)		Trauma (*n* = 61)			ADUPs and Trauma (*n* = 66)			None (*n* = 106)			*F/H/χ* ^2^
	*M* (*SD*)	*%*	*M* (*SD*)		*%*	*M* (*SD*)		*%*	*M* (*SD*)		*%*	
Age	39.9 (10.6)		41.2 (12.9)			38.6 (11.9)			43.2 (10.1)			2.59
Annual income ^1^	4.67 (2.36)		4.35 (2.21)			4.24 (2.21)			4.87 (2.43)			3.13
Origin												4.24
Spain		82.3			69.9			81.7			75.6	
Latin America		8.90			6.70			6.10			7.50	
Europe (except Spain)		3.80			20.1			6.10			11.2	
Africa		3.80			3.30			6.10			4.80	
Asia		1.20			-			-			0.90	
Educational level												13.3
No education		10.1			6.70			13.6			4.70	
Primary		50.6			51.7			48.5			43.4	
Secondary		30.4			40.0			33.3			39.6	
University		8.90			1.60			4.60			12.3	
Marital status												10.4
Married or with partner		24.1			23.3			22.7			21.7	
Single		36.7			38.3			47.0			33.9	
Separated		7.60			10.1			10.6			14.2	
Divorced		31.6			28.3			18.2			30.2	
Widowed		-			-			1.50			-	
Employment												10.8
Yes		68.4			53.3			54.5			69.8	
No		31.6			46.7			45.5			30.2	

*Note.* All comparisons were not significant at the 0.05 level. ADUPs = Alcohol and/or other Drug Use Problems, *M* = Mean, *SD* = Standard Deviation, *F* = one-way ANOVA, *H* = Kruskal–Wallis H test, *χ*^2^ = chi-square test. ^1^ Annual income: 1 = <EUR 1800, 2 = EUR 1800–EUR 3600, 3 = EUR 3600–EUR 6000, 4 = EUR 6000–EUR 12,000, 5 = EUR 12,000–EUR 18,000, 6 = EUR 18,000–EUR 24,000, 7 = EUR 24,000–EUR 30,000, 8 = EUR 30,000–EUR 36,000, 9 = EUR 36,000–EUR 60,000, 10 = EUR 60,000–EUR 90,000, 11 = EUR 90,000–EUR 120,000, and 12 = >EUR 120,000.

**Table 3 behavsci-15-00156-t003:** Comparison of risk of intimate partner violence and variables related to emotion regulation by group.

Variables	ADUPs (*n* = 79)	Trauma (*n* = 61)	ADUPs and Trauma (*n* = 66)	None (*n* = 106)	Group Effect	
	*M* (*SD*)	*M* (*SD*)	*M* (*SD*)	*M* (*SD*)	*F*	*η* ^2^ * _p_ *
Risk of IPV recidivism	11.2 (4.23)	12.3 (4.19)	15.1 (5.36)	8.37 (3.65)	33.1 **	0.26
Physical violence	1.89 (5.40)	1.17 (4.48)	5.23 (9.62)	1.27 (4.88)	6.51 **	0.06
Psychological violence	6.59 (10.4)	6.77 (9.50)	10.9 (12.1)	3.88 (7.64)	7.05 **	0.07
Depressive Symptomatology	2.08 (0.81)	2.13 (0.93)	2.26 (0.85)	1.73 (0.76)	6.67 **	0.06
Difficulties in identifying feelings	2.46 (1.29)	2.34 (1.19)	2.92 (1.26)	2.01 (1.05)	8.01 **	0.07
Difficulties in describing feelings	2.82 (1.24)	2.84 (1.17)	3.25 (1.19)	2.55 (1.16)	4.55 **	0.04
Anxiety	53.7 (34.1)	50.8 (36.5)	68.5 (28.9)	34.8 (32.6)	13.7 **	0.13
Dysthymia	32.2 (26.5)	32.3 (30.6)	44.7 (28.4)	20.3 (26.1)	10.1 **	0.09
Post-traumatic stress disorder	37.9 (27.8)	39.8 (28.4)	52.7 (25.8)	20.9 (25.8)	18.8 **	0.16

*Note.* ADUPs = Alcohol and/or other Drug Use Problems, IPV = Intimate Partner Violence, *M* = Mean, *SD* = Standard Deviation. ** *p* < 0.01.

**Table 4 behavsci-15-00156-t004:** Comparison of the risk of intimate partner violence and variables related to emotion regulation over time by group.

Variables	Pre-Treatment				Post-Treatment				Time Effect		Time-Group Effect	
	ADUPs (*n* = 56)	Trauma (*n* = 38)	ADUPs and Trauma (*n* = 39)	None (*n* = 87)	ADUPs (*n* = 56)	Trauma (*n* = 38)	ADUPs and Trauma (*n* = 39)	None (*n* = 87)				
	*M* (*SD*)	*M* (*SD*)	*M* (*SD*)	*M* (*SD*)	*M* (*SD*)	*M* (*SD*)	*M* (*SD*)	*M* (*SD*)	*F*	*η* ^2^ * _p_ *	*F*/*χ*^2^	*η*^2^*_p_*/*V*
SARA	11.0 (4.22)	11.6 (3.84)	15.0 (5.35)	8.11 (3.33)	8.13 (2.86)	9.24 (2.96)	10.8 (3.79)	6.59 (2.58)	166 **	0.44	7.61 **	0.10
CTS Ph	1.80 (4.98)	1.37 (5.32)	5.69 (9.85)	1.43 (5.31)	1.13 (4.66)	1.18 (2.65)	3.31 (7.37)	1.28 (5.31)	2.76	0.01	0.95	0.01
CTS Ps	7.17 (11.1)	7.50 (10.6)	14.1 (12.3)	3.80 (7.48)	3.96 (8.29)	5.34 (9.05)	7.26 (10.8)	4.40 (9.11)	12.3 **	0.05	3.92 **	0.05
CES-D	2.04 (0.82)	2.27 (0.97)	2.22 (0.79)	1.72 (0.73)	1.77 (0.66)	1.95 (0.82)	2.03 (0.78)	1.57 (0.59)	16.9 **	0.07	0.54	<0.01
TAS-20 I	2.52 (1.29)	2.51 (1.28)	2.90 (1.32)	2.05 (1.09)	2.39 (1.17)	2.24 (1.05)	2.45 (1.23)	1.88 (0.91)	12.9 **	0.06	1.00	0.01
TAS-20 D	2.83 (1.27)	2.92 (1.15)	3.26 (1.17)	2.58 (1.19)	2.79 (1.08)	2.67 (0.99)	2.94 (0.92)	2.49 (1.02)	4.85 *	0.02	0.71	0.01
Dropout ^1^											8.63 *	0.17
No					57 (72.2)	43 (70.5)	41 (62.1)	87 (82.1)				
Yes					22 (27.8)	18 (29.5)	25 (37.9)	19 (17.9)				

*Note.* ADUPs = Alcohol and/or other Drug Use Problems, *M* = Mean, *SD* = Standard Deviation, SARA = Spousal Assault Risk Assessment, CTS Ph = Conflict Tactics Scale—physical violence, CTS Ps = Conflict Tactics Scale—psychological violence, CES-D = Center for Epidemiologic Studies Depression Scale, TAS-20 I = Toronto Alexithymia Scale—20 Items—Identifying feelings, TAS-20 D = Toronto Alexithymia Scale—20 Items—Describing feelings. ^1^ For dropout data, *n* and % are provided. * *p* < 0.05, ** *p* < 0.01.

## Data Availability

Research data are not shared for ethical reasons.

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
