# Peer review of "The Impact of Trauma and Substance Use on Emotion Regulation and Intimate Partner Violence Perpetration: Implications for Perpetrator Programs"

_behavsci, 2025, doi:10.3390/bs15020156_

Round 1
Reviewer 1 Report
Comments and Suggestions for Authors
This study examined trauma, substance use, and emotion regulation in a sample of domestic abusers in a treatment program. Findings from this sample of 312 men indicated that men who had both trauma and substance use problems had more emotion regulation problems, IPV and recidivism, dropout, and were most treatment resistant.
Overall, the manuscript was well-written, the sample is fairly large from several programs, and the assessment strategy was sound. My concerns are with the unique contribution of this study, the lack of review of relevant background material, and methodological issues.
Perhaps most importantly, it is not clear exactly how this study advances our current state of knowledge. Findings generally replicated what prior studies have shown regarding the impact of trauma and substance use on IPV and emotion regulation.
Relatedly, there is other research that the authors did not review regarding trauma informed abuser intervention programs that reduced alexithymia and abusive behaviors. There are a number of studies on trauma informed abuser intervention and the link between trauma, PTSD, and abuse that were not included in the literature review.
My largest methodological concern is a lack of rationale for the different comparison groups. The study was largely atheoretical such that there was not a guiding model or theory that guided comparisons between groups. Findings were generally what one might expect; those with greater problems demonstrated poorer outcomes, but we do not know the mechanisms for these findings.
Most of the measures classified as emotion regulation variables do not seem to be truly measuring emotion regulation.
While data showed that abuser intervention reduced emotion regulation problems, not enough information is provided about the intervention to know the mechanisms for such changes.
Reviewer 2 Report
Comments and Suggestions for Authors
This study aimed to examine how childhood and/or adolescent trauma and substance misuse among court-mandated participants in an IPV perpetrator program influence dropout, emotion regulation, and IPV. Additionally, authors aim to assess a potential need for specialized interventions. This study contributes to existing literature by providing evidence for the relationship between ADUPs and childhood/adolescent trauma, and higher emotional dysregulation, IPV at intake, and dropout rates. Best models for programming are noted. Measures have acceptable to good reliability. Overall, the manuscript is definitely relevant for the field and presented in a well-structured manner. Tables and figures are clean and accurate.
This manuscript is ambitious, although lacks important depth for clear understanding, particularly in the form of consideration for controls, operationalization, and conceptual flow. Claims are made about a lack of research and emerging studies, but there is literature evidence. Other research studies are briefly referenced in ways to supporting study pursuits, not for critical analysis of methodology or as a foundation for further research. Further, the introduction begins with an emphasis on best practice for program effectiveness and need for evidence-based emotion regulation strategies, which will seemingly lead to an evaluation of the program intervention. Yet there is no mention about the specifics of the program intervention model or content of the perpetrator program studied. The focus shifts toward needing to identify specific risk factors and variables to target, yet the following sentence names alcohol and other drug use problems and trauma as prominent risk factors. A testable hypothesis is included, although a bit wordy and unclear, which is more related to sentence structure and grammar than scientific content. In terms of controls, does this program follow a cohort model or open to new members joining groups continuously? If the latter, how does length of time that a participant has been participating in the program influence findings? Unfortunately, the ADUP measure is behind a paywall. Does ADUPs refer to current and/or past behavior or ADUP at the time of the IPV? It would be helpful to clarify from the beginning that the trauma being studied is specific to childhood and adolescence. The measure of trauma appears to be based on a dichotomous “yes” or “no,” response if so, how does post-traumatic growth factor?
The multiple alpha coefficients below acceptability are not acknowledged or addressed as such. A Cronbach alpha coefficient is not listed for Multiaxial Inventory-III. There is no reference to missing data if any. Statistical analyses are said to be conducted using an outdated version of SPSS (28) instead of SPSS V 29.
Ethics statements and data availability statements are adequate. Of concern, the study sample used only court-mandated clients. The introduction references suspended sentences contingent upon participation in community-based intervention programs. Does this apply to the specific programming studied in this research? The court-mandated status is not addressed as a limitation or influence on dropout rate. In other words, did participants drop out because they were reincarcerated? While noted that participants were informed that declining to participate would not affect their legal status (line 212), was it made clear that opting to participate would also not benefit their legal status?
Cited references do not appear to be of the most recent publications. The majority of publications fall outside the last 5 years time frame. The first four authors all have few self-citations. The fifth author self-cites 12 times. A significant portion of the discussion section is based on another study, seemingly from the same data set by the first author. This data set seems to be related to other recently published articles (DOI 10.5093/pi2024a13; DOI: 10.1080/15564886.2024.2322960). Due to a paywall, it is unclear if the results of this manuscript were already submitted or published in part.
Specifics
Line 10 The terminology “Alcohol and/or other drug use problems (ADUPs)" is unfamiliar to me. How does this differ from substance misuse?
Line 14: At this point in the manuscript, “drop out” is unclear. Add clarification of drop out as it relates to the perpetrator program as opposed to study participation.
Line 15: Similar to line 14, at this point in the manuscript it is unclear what IPV means. Please clarify risk of IPV recidivism vs. actual recidivism rates.
Line 21: By intake do you mean when they began the perpetrator program or pre-intervention in study?
Line 44: What model does this intervention use? Risk-Needs-Responsivity (RNR) model? Principles of Effective Intervention? Neither? How do authors know this intervention is not following these models?
Line 117. Does this mean perpetrators were in these groups for four years or is the sample of different intervention groups? Did they have the same group facilitators? Is the intervention manualized?
Line 146. Is the self-report nature of IPV a limitation? Might someone want to avoid self incrimination especially if this program is connected to the criminal justice system?
Line 203: As in the program intake or assessment with researchers?
Line 229. Was normality and homogeneity of variances assessed for ANOVA?
The quality of English did not significantly interfere with understanding, although did at times create some confusion.
Reviewer 3 Report
Comments and Suggestions for Authors
I would like to thank the Editor for the opportunity to review this manuscript and the authors for their work.
The manuscript focuses on the risk given by exposure to trauma and substance use on a number of variables related to the perpetration of IPV and emotional regulation, providing interesting data useful at the clinical and intervention levels.
Below are my revisions:
- Line 15: What intervention are you talking about? Specify it from the abstract
- lines 40-41: elaborate and describe in more detail the findings of the cited systematic reviews
- line 109: Why was it decided to consider depressive symptoms and clinical syndromes as indicators of emotional regulation? The rationale for this choice should be clarified
- line 115: Please provide more detail on the characteristics of the intervention? What kind of intervention is it? What is the theoretical approach and mode of delivery of the intervention? On what is it focused? What techniques does it use? How long did it last?
- Line 253: When you talk about baseline characteristics compared across groups, what variables are you talking about specifically? Please clarify that are sociodemographic variables and specify them.
Round 2
Reviewer 1 Report
Comments and Suggestions for Authors
The authors were very responsive to my previous concerns and the manuscript is very much improved. I have no remaining concerns.
Author Response
We sincerely appreciate your thoughtful review and your kind acknowledgment of the improvements made to the manuscript. Your feedback greatly contributed to enhancing the quality and clarity of our work. Thank you for your time and effort throughout the review process.